# Characterization and Genome Analysis of *Cladobotryum mycophilum*, the Causal Agent of Cobweb Disease of *Morchella sextelata* in China

**DOI:** 10.3390/jof9040411

**Published:** 2023-03-27

**Authors:** Zhenghui Liu, Yunlong Cong, Frederick Leo Sossah, Yongzhong Lu, Jichuan Kang, Yu Li

**Affiliations:** 1Engineering and Research Center for Southwest Bio-pharmaceutical Resources of National Education Ministry, Guizhou University, Guiyang 550025, China; 2Engineering Research Center of Edible and Medicinal Fungi, Ministry of Education, Jilin Agricultural University, Changchun 130118, China; 3Research Institute of Science and Technology, Guizhou University, Guiyang 550025, China; 4Council for Scientific and Industrial Research (CSIR), Oil Palm Research Institute, Coconut Research Programme, Sekondi P.O. Box 245, Ghana; 5School of Food and Pharmaceutical Engineering, Guizhou Institute of Technology, Guiyang 550003, China

**Keywords:** *Morchella sextelata*, cobweb disease, identification, *Cladobotryum mycophilum*, whole genome sequencing

## Abstract

Cobweb disease is a fungal disease that can cause serious damage to edible mushrooms worldwide. To investigate cobweb disease in *Morchella sextelata* in Guizhou Province, China, we isolated and purified the pathogen responsible for the disease. Through morphological and molecular identification and pathogenicity testing on infected *M*. *sextelata*, we identified *Cladobotryum mycophilum* as the cause of cobweb disease in this region. This is the first known occurrence of this pathogen causing cobweb disease in *M*. *sextelata* anywhere in the world. We then obtained the genome of *C*. *mycophilum* BJWN07 using the HiFi sequencing platform, resulting in a high-quality genome assembly with a size of 38.56 Mb, 10 contigs, and a GC content of 47.84%. We annotated 8428 protein-coding genes in the genome, including many secreted proteins, host interaction-related genes, and carbohydrate-active enzymes (CAZymes) related to the pathogenesis of the disease. Our findings shed new light on the pathogenesis of *C*. *mycophilum* and provide a theoretical basis for developing potential prevention and control strategies for cobweb disease.

## 1. Introduction

Morels (*Morchella* spp.) are a rare edible and medicinal mushroom known for their high nutritional value, including high protein and low-fat content, and rich variety of essential minerals, vitamins, and other nutrients. They also have antioxidant, anti-tumor, and immune-regulating properties, making them a promising source of medicinal benefits with broad development prospects [1,2,3,4,5,6].

Wild morels are mainly found in Yunnan, Sichuan, Gansu, Heilongjiang, and Xinjiang provinces in China [7]. Morels have been cultivated for over 130 years, with the cultivation area expanding from 6.67 ha in 2003 to 10,050 ha in 2020 [7]. As of today, morels are cultivated almost everywhere in China, except for Hainan, Taiwan, Hong Kong, and Macao, and have become an important tool for poverty alleviation and rural revitalization [8].

In recent years, the cultivation and scale of morels have increased, but this has also led to new challenges. These challenges include changing and abnormal weather patterns and disease problems resulting from improper cultivation management. These factors have become important reasons for serious production losses or crop failure [9], which significantly hinder the stable development of the morel industry. Currently, reported morel diseases mainly include white mold disease caused by *Paecilomyces penicillatus* infection [9], handle rot disease caused by *Fusarium* sp. infection [10], rot disease caused by *Lecanicillium aphanocladii* infection [11], and cobweb disease caused by *Cladobotryum protrusum* infection [12]. These diseases have greatly reduced the economic benefits of the morel industry.

Cobweb disease is caused by the *Cladobotryum* genus, which is a significant threat to edible mushroom cultivation worldwide [13]. This disease can cause production losses of up to 40% [14] and is considered one of the most destructive diseases of edible mushrooms [15]. Different species of *Cladobotryum* can infect a wide range of cultivated edible fungi, causing varying degrees of damage. For example, *C*. *dendroides* mainly infects *Lentinus edodes* [16] and *Agaricus bisporus* [17], while *C*. *varium* (teleomorph sexuales: *Hypomyces aurantius*) mainly infects *Hypsizygus marmoreus* [18], *Flammulina filiformis* [19], and *Oudemansiella raphanipies* [20]. *C*. *semicirculare* mainly infects *Ganoderma tsugae* [21], *C*. *cubitense* mainly infects *Auricularia polytricha* [22], and *C*. *protrusum* mainly infects *Coprinus comatus* [23] and *M*. *importuna* [12]. *C*. *mycophilum* (teleomorph sexuales: *H*. *odoratus*) has been reported to infect a relatively wide range of edible mushrooms, including *A*. *bisporus* [24], *G*. *lucidum* [25], and *Pleurotus eryngii* [26,27]. Although the pathogen responsible for cobweb disease in *M*. *importuna* in Shandong Province was identified as *C*. *protrusum* [12], our investigation of the incidence of cobweb disease in *M*. *sextelata* in Guizhou suggests that there may be more than one pathogen causing this disease.

High-throughput sequencing technology has become more prevalent in recent years due to its rapid development and decreasing costs. This has led to increased exploration and study of fungal genomic information. Recently, the genomes of two species of *Cladobotryum*, *C*. *protrusum* and *C*. *dendroides* [28,29], commonly responsible for causing cobweb disease in edible mushrooms, have been sequenced. Analysis of these genomes has revealed a large number of pathogenic genes, including those involved in peptidase, carbohydrate-active enzyme, cytochrome P450 enzyme, and secondary metabolites such as mycotoxins and pigments [28,29]. These genomic findings have provided valuable insights into identifying genes related to growth, evolution, host-pathogen interactions, and pathogenicity in *Cladobotryum* fungi.

This study aimed to identify the pathogen responsible for cobweb disease in *M*. *sextelata* in Guizhou Province, China. We identified the pathogen of cobweb disease as *C*. *mycophilum* and sequenced its whole genome using the HiFi sequencing platform. This was the first time the genome of *C*. *mycophilum* had been sequenced. The main objectives of the study were to identify the pathogen causing cobweb disease and to provide a high-quality reference genome for comparative genomic studies of the fungal family Hypocreaceae and other mycoparasites. Additionally, we analyzed the genes responsible for the pathogenicity and fungal parasitism of *C*. *mycophilum*. The high-quality genome assembly of *C*. *mycophilum* will expand the genome database and aid in comparative genomic studies. This study’s findings will help analyze how cobweb disease develops and spreads, leading to more effective control strategies against the disease.

## 2. Materials and Methods

### 2.1. Field Surveys

In June 2022, a field survey was conducted to investigate the incidence and symptoms of cobweb disease in *M. sextelata* during the off-season at a forest cultivation base in Weining County, Bijie City, Guizhou Province, China. A total of 300 *M. sextelata* fruiting bodies were investigated in triplicate, with 100 fruiting bodies being studied in different *M*. *sextelata* arch sheds. Additionally, 10 *M*. *sextelata* fruiting bodies with typical cobweb disease symptoms were collected for further studies. The aim was to determine the incidence and symptoms of the disease and disease occurrence period as well as to isolate and culture the pathogen responsible for the disease.

### 2.2. Isolation and Purification of the Fungal Pathogens

To isolate and purify the fungal pathogens responsible for cobweb disease in *M*. *sextelata*, small tissue blocks were taken from the healthy junction of new fruiting bodies displaying cobweb disease symptoms. Before inoculating these tissue blocks onto PDA plates, they were disinfected with a 75% ethanol solution for 30 s, followed by a 1% sodium hypochlorite solution, and washed three times with sterile water. The plates were then cultured in a dark 25 °C incubator. Once the mycelium of the fungi had grown, single spores were isolated from these cultures to obtain pure cultures of the pathogen.

### 2.3. Morphological and Molecular Biological Characterization of Pathogens

#### 2.3.1. Morphological Identification

The purified pathogens were inoculated on PDA plates and cultured in the dark at 25 °C for 5, 10, and 20 days to observe the morphological characteristics of the fungal colony, including the appearance of the mycelium, conidiophores, conidia, and chlamydospores. In addition, the sizes of 60 conidia were measured. The morphological identification of the fungal pathogens followed the method described in a book titled “*The Genera of Hyphomycetes*” [30].

#### 2.3.2. Molecular Biology Identification

Genomic DNA was extracted using the novel plant genomic DNA extraction kit NuClean Plant Genomic DNA Kit (CWBIO, Beijing, China), and the genome extraction quality was determined by 1% agarose gel electrophoresis. Primers ITS4 and ITS5 [31], EF1-983f [32] and EF1-2218r [33], and RPB2-5f and RPB2-7cR [34] were used for PCR amplification. PCR amplifying system (25 μL): ddH_2_O 14.5 μL, 5 μL 5 × TranStart Faspfu Buffer, dNTPs (2.5 mM) 2 μL, 1 μL forward primer, 1 μL reverse primer, 1 μL genomic DNA, and 0.5 μL TranStart Faspfu DNA Polymerase (5 U/μL). PCR reaction conditions: 95 °C pre-denaturation for 5 min, followed by 95 °C denaturation for 30 s, 55 °C (ITS), 59 °C (TEF1), and 58 °C (RPB2) annealing for 30 s, 72 °C extension for 1 min, a total of 30 cycles, and finally 72 °C extension for 5 min. The PCR amplification products were confirmed by 1% agarose gel electrophoresis and then sent to Beijing Qingke Biotechnology Co., Ltd. (Beijing, China) for sequencing. 

The sequencing results were subjected to BLAST sequence analysis in GenBank (NCBI, http://www.ncbi.nlm.gov (accessed on 12 November 2022)), the correct sequences of the same and similar species as the pathogens were selected from GenBank and downloaded, and the multiple sequence comparison was performed by BioEdit v7.2.5 software [35]. Phylogenetic analyses were performed based on ITS, TEF1, and RPB2 sequence data. Maximum likelihood analysis (ML) was performed using the GTR + G + I model of RAxML-8.0.26 [36], and Bayesian inference (BI) analysis was performed using MrBayes v3.2 [37] to determine the posterior probability (PP). The identification of the pathogen strains was determined according to the phylogenetic relationship. The gene sequences used to construct the phylogenetic tree are shown in Table 1.

### 2.4. Pathogenicity Determination

To determine the pathogenicity of the isolated and purified pathogen strains, they were grown on PDA plates in a constant temperature dark environment at 25 °C for 15 days. After collecting the conidia, a spore suspension was prepared by diluting them with sterile water to a concentration of 5 × 10^6^ spores/mL. This suspension was sprayed onto the foundation soil of healthy *M*. *sextelata* fruiting bodies in a controlled cultivation shed with a temperature range of 10~18 °C, and the relative humidity was 85~95%.

Nine fresh and healthy *M*. *sextelata* fruiting bodies were inoculated with a 10-µL droplet of the conidial suspension of each pathogen strain. In contrast, fresh *M*. *sextelata* fruiting bodies inoculated with sterile water were used as controls for each experiment. The disease incidence was observed and recorded daily in both inoculated and uninoculated fruiting bodies. The pathogen was reisolated from new symptomatic *M*. *sextelata* fruiting bodies to fulfil Koch’s postulate. This pathogenicity test was repeated three times.

### 2.5. Genome Sequencing and Assembly

The genomic DNA of *C*. *mycophilum* BJWN07 was extracted using the SDS method [38]. The DNA was assessed for quantity, quality, and integrity using agarose gel electrophoresis, Qubit^®^ 2.0 Fluorometer (Thermo Fisher Scientific, Foster City, CA, USA), and Agilent 2100 bioanalyzer (Agilent Technologies, Santa Clara, CA, USA). For PacBio sequencing, the DNA was sheared with Covaris g-TUBE (Covaris, MA, USA) into target fragment size, and DNA damage and fragments were repaired. The DNA fragment was then purified and selected using AMpure PB magnetic beads (PacBio, CA, USA) to construct the SMRT Bell Library. The BluePippin system (SageScience, MA, USA) was used to select an insert size of 20 kb. The SMRT Bell library was sequenced on the PacBio RSII platform (Pacific Biosciences, Menlo Park, CA, USA).

For Illumina sequencing, the DNA library was constructed with an insert size of 350 bp using the NEBNext^®^ Ultra™ DNA Library Prep Kit for Illumina (NEB, Ipswich, MA, USA). The sequencing libraries were analyzed using the Agilent 2100 Bioanalyzer (Agilent Technologies, Santa Clara, CA, USA). Once the library inspection was qualified, the whole genome was sequenced on the Illumina HiSeq PE150 (Illumina, San Diego, CA, USA). High-precision HiFi reads were generated and assembled using the CCS software. The polished consensus sequences were corrected using Illumina sequencing data for the final assembly. All sequencing and library preparation was carried out at the Beijing Novogene Bioinformatics Technology Co., Ltd. (Beijing, China).

### 2.6. Genome Component Prediction

We used seven databases GO (Gene Ontology) [39], KEGG (Kyoto Encyclopedia of Genes and Genomes) [40,41], KOG (Clusters of Orthologous Groups) [42], NR (Non-Redundant Protein Database databases) [43], TCDB (Transporter Classification Database) [44], P450, and Swiss-Prot [45] to predict gene functions. We performed a whole genome BLAST search with an E-value lower than 1e-5 and a minimum alignment length percentage larger than 40% against these seven databases. Additionally, we used Signal P [46] to predict secretory proteins and the antiSMASH tool [47] to analyze secondary metabolism gene clusters. For pathogenic fungi, we added pathogenicity using the PHI (Pathogen Host Interactions) [48] and DFVF (database of fungal virulence factors databases). The carbohydrate-active enzymes were predicted using the dbCAN web server (https://bcb.unl.edu/dbCAN2/) (accessed on 16 November 2022) [49].

### 2.7. Phylogenomics Analysis of C. mycophilum

The protein-coding genes from 14 species, including *C. mycophilum*, *H. rosellus* [29], *C. protrusum* [28], *H. perniciosus* [50], *Trichoderma longibrachiatum* [51], *Neurospora crassa* [52], *Magnaporthe grisea* [53], *Pyricularia oryzae* [54], *Fusarium solani*, *F. oxysporum* [55], *Tolypocladium inflatum* [56], *T. virens* [57], *T. reesei* [58], and *Clonostachys rosea* [59] (Appendix A), were compared using BLASTP with an E-value cutoff of 1 × 10^−5^. The OrthoMCL (v.2.0.9) [60] method was then used to identify direct orthologous groups. Maximum likelihood (ML) analysis was performed using RAxML-8.0.26 [36] with 1000 bootstrap repeats.

## 3. Results

### 3.1. Cobweb Disease Symptoms and Incidence 

Cobweb disease symptoms were observed in a morel farm located in Weining County, Bijie City, Guizhou Province, China. The disease incidence ranged from 5% to 60%. The most common symptoms observed were a small amount of white flocculate aerial mycelium on the surface of the stipe (Figure 1A), thick white mycelium covering the fruiting body (resembling cotton catkins), and the fruiting body becoming soft (Figure 1B). In severe infections, the entire fruiting body was covered with white hyphelium, the stipe was lodged, and, eventually, the whole mushroom died (Figure 1C). During the later stage of the infection, the pathogens produced numerous conidia, and the mycelium changed from white to pink (Figure 1D).

### 3.2. Identification of Pathogen

#### 3.2.1. Morphological Characterization of the Pathogen

To begin with, strains BJWN07, BJWN12, and BJWN24 showed consistent morphological and micromorphological features. These strains grew rapidly on PDA plates after 5 days of cultivation and formed colonies that reached 60–75 cm in diameter with abundant aerial hyphae that resembled cotton wool (Figure 2A,B). Initially, the colonies were white, but over time they turned yellow from the center and became progressively darker, reaching a dark yellow color after 25 days (Figure 2C,D), pink after 50 days (Figure 2E,F), and dark pink after 60 days (Figure 2G,H). The conidial stem had a septum and branches that were either broom or round in shape (Figure 2I,J). The fungus produced chlamydospores, and cells expanded into a string (Figure 2K). The conidia were colorless, oval-shaped, blunt round at both ends, and measured 17.2–19.8 × 8.4–9.3 μm. They had 1–3 diaphragms and were slightly bent at the diaphragm (Figure 2L–O).

#### 3.2.2. Molecular Identification of the Pathogen

The internal transcribed spacer (ITS), TEF1, and RNA polymerase II largest subunit (RPB2) regions of three strains, BJWN07, BJWN12, and BJWN24, were amplified by PCR and sequenced. The ITS fragment length was between 562–566 bp (GenBank accession numbers OP714368, OP714369, and OP714393), the TEF1 fragment length was between 927–939 bp (GenBank accession numbers OP759638, OP759639, and OP759640), and the RPB2 fragment length was between 1118–1121 bp (GenBank accession numbers OP718561, OP718562, and OP718563). After searching and comparing the sequences with the NCBI database BLAST, it was found that they were 100% similar to the accession numbers MH185858 (ITS), HF911622 (TEF1), and OK458561 (RPB2), respectively.

Next, a phylogenetic tree was constructed using the rDNA ITS regions and partial sequences of TEF1 and RPB2 genes. The sequences of strains BJWN07, BJWN12, and BJWN24 were found to be in the same branch as *H*. *odoratus* (asexual type: *C*. *mycophilum*) (Figure 3) and were most closely related with high statistical support (ML/BI: 100/1). The strains BJWN07, BJWN12, and BJWN24 were identified as *C. mycophilum* based on morphological and molecular phylogenetic analyses.

### 3.3. Pathogenicity Determination

The pathogenicity of three strains (BJWN07, BJWN12, and BJWN24) was investigated by inoculating healthy fresh soil with spore suspensions at the base of the stipe (Figure 1E). After seven days, white, cobweb-like mycelia appeared, infecting the stipe’s base and gradually spreading to the cap (Figure 1F). The fruiting body became soft, and the mycelium covered it entirely, leading to lodging and death (Figure 1G). These symptoms were consistent with those observed under natural conditions. Inoculation with sterile water had no effect (Figure 1H). Koch’s postulates were verified, and molecular identification confirmed the presence of the same pathogen.

### 3.4. Genome Sequencing and Assembly

The representative strain BJWN07 was sequenced using the HiFi sequencing platform, generating a total of 4.91 Gb of data with a sequencing depth of approximately 130×. The assembled genome is 38.56 Mb, consisting of 10 contigs, with a GC content of 47.84% and an N50 value of 5.42 Mb. The genome assembly resulted in the identification of 8428 protein-coding sequences (CDS) and 330 RNA genes, including 264 tRNA, 53 5S rRNA, 6 18S rRNA, and 7 28S rRNA genes (Table 2). The prediction of repeated DNA sequences in the BJWN07 genome identified 2198 long terminal repeats, which make up 0.4237% of the genome, with a total length of 171,302 bp. Additionally, there were 2239 DNA transposons that constituted 0.7662% of the genome, with a total length of 309,760 bp. The genome also contained 977 scattered repeats that added up to 83,955 bp, representing 0.2077% of the genome. Among these scattered repeats were 47 short scattered repeats, 104 rolling rings, and 70 unknown scattered repeats. Tandem repeat prediction identified 11,990 tandem repeats with a total length of 492,550 bp, making up 1.2183% of the genome. The genome also contained 7609 small-satellite DNA sequences, which added up to 311,395 bp, representing 0.7702% of the genome. In addition, there were 3071 microsatellite DNA sequences with a total length of 118,888 bp, representing 0.2941% of the genome.

### 3.5. Gene Function Annotation

A total of 8428 protein-coding genes were predicted. The protein-coding genes had an average length of 11.43 Mb, representing 29.64% of the total gene length. These protein-coding genes were annotated in various databases such as GO, KEGG, KOG, NR, Pfam, Swiss-Prot, TCDB, CAZy, Secretory_Protein, cytochrome P450, PHI, and DFVF databases (Table 3). Among these databases, the Nr database annotated 7766 protein sequences, which had the closest matches to *Escovopsis weberi* (1344), *T*. *arundinaceum* (1182), *T*. *harzianum* (572), *T*. *virens* (453), and *T*. *asperellums* (419) (Appendix A).

The GO functional category predicted a total of 5683 genes, which accounted for 67% of the total predicted genes (Appendix A). The top 10 most abundant function categories were “catalytic activity”, “metabolic process”, “binding”, “cellular process”, “cell”, “cell part”, “localization”, “establishment of localization”, “organelle”, and “biological regulation” (Appendix A).

The KEGG database was used to map the predicted genes, and 7447 gene models (87.80% of the total number of genes) were functionally classified. Several categories related to metabolism and membrane transport were highly enriched, including “Global and overview maps” (876), “Translation” (298), “Carbohydrate metabolism” (278), “Transport and catabolism” (269), “Amino acid metabolism” (246), “Signal transduction” (232), and “Folding, sorting and degradation” (223) (Appendix A).

In the KOG category, there were 1964 genes (Appendix A). The category with the highest number of genes was “Posttranslational modification, protein turnover, chaperones” with 222 genes, followed by “General function prediction only” with 210 genes, “Translation, ribosomal structure and biogenesis” with 208 genes, “Energy production and conversion” with 166 genes, and “Amino acid transport and metabolism” with 166 genes.

The TCDB database annotation assigned 551 protein-coding genes to seven functional categories, including “Channels/pores”, “Electrochemical Potential-driven Transporters”, “Primary Active Transporters”, “Group Translocators”, “Transmembrane Electron Carriers”, “Accessory Factors Involved in Transport”, and “Incompletely Characterized Transport Systems” (Figure 4). The categories with the highest number of genes were “Electrochemical Potential-driven Transporters” with 176 genes and “Primary Active Transporters” with 160 genes.

The BJWN07 genome contained 499 annotated CAZymes, which included various families of carbohydrate-binding molecules (CBM), carbohydrate esterases (CE), glycoside hydrolases (GHs), glycosyltransferases (GTs), polysaccharide lyases (PLs), and auxiliary activities (AA) (Table 4). The GHs family had the most genes, with 249 genes accounting for 49.90% of the total number of CAZymes. The next largest family was the GTs family, with 111 genes accounting for 22.16% of the total number of CAZymes. Among the GHs families, GH18 had the highest number of genes at 31, while in the GTs family, GT31 had the highest number of genes at 15.

The antiSMASH analyses identified 78 secondary metabolic gene clusters and 773 secondary metabolic genes (Table 5). Of these, 23 clusters were identified as type 1 polyketide synthase (T1PKS) gene clusters containing 229 genes, accounting for 29.62% of the total secondary metabolic genes. The analysis also identified 18 non-ribosomal peptide synthetase (NRPS) gene clusters containing 175 genes, representing 22.64% of the total secondary metabolic genes. Additionally, 12 terpene gene clusters were identified, containing 59 genes. There were eight NRPS-T1PKS gene clusters containing 79 genes and seven NRPS-like gene clusters containing 69 genes. Overall, this analysis provides insight into the metabolic potential of *C*. *mycophilum* BJWN07 and the types of secondary metabolites it may produce.

Annotation of the genome by the PHI database identified 1429 proteins related to pathogenicity, representing 16.85% of all the encoded proteins in the genome (Figure 5). Among these, the most abundant category is “reduced virulence,” which includes 567 proteins and accounts for 39.68% of the total candidate pathogenicity-related proteins. The second most abundant category is “unaffected pathogenicity,” which includes 535 proteins and accounts for 37.44% of the total candidate pathogenicity-related proteins. Additionally, 104 proteins belong to the “loss of pathogenicity” category, and 84 proteins belong to the “lethal” category.

### 3.6. Phylogenomics Analysis of C. mycophilum

From the 14 species, a total of 16,159 direct orthologous groups were identified. To construct a phylogenetic tree, 1731 single-copy orthologous genes were used. The analysis revealed that *C*. *mycophilum* BJWN07, *H*. *rosellus* (asexual type was *C. dendroides*), and *C*. *protrusum* belong to the same genus. However, *C*. *mycophilum* BJWN07 is distantly related to the outgroup *N. crassa* (Figure 6).

## 4. Discussion

In this study, we investigated cobweb disease affecting *M*. *sextelata* in Weining County, Bijie City, Guizhou Province, China. Our findings revealed that the disease had a median incidence of 15%. The causal agent was identified as *C*. *mycophilum*, which was previously unreported as causing cobweb disease in *M. sextelata* in China or globally. Cobweb disease, caused by different *Cladobotryum* species [61,62,63], has been observed in many countries, including China [12,22,25,64,65], Korea [26,63], South Africa [24], and Spain [27,66,67], causing significant economic losses in the edible mushroom industry [13,15]. *C. mycophilum* has a broad host range and produces a large number of conidia in the form of dry powder in the later stages of infection, making it easily transmitted by wind and humidity, resulting in outbreaks and epidemics [26,27,61,62,63]. Due to the expansion of the morel industry and inadequate adoption of good agricultural practices, cobweb disease has become a significant challenge in various cultivation areas in Guizhou Province. Therefore, we recommend a comprehensive investigation of cobweb disease in the primary morel cultivation areas in China, particularly Guizhou Province, to identify pathogen species and their genetic diversity in different regions. This approach may provide a theoretical basis for the scientific prevention and control of cobweb disease, which is crucial for sustaining the morel industry.

The genome size of *C. mycophilum* assembled in this study is 38.56 Mb, which is the first high-quality genome of *C. mycophilum* published and similar to those of the other two species of the same genus reported previously viz *C*. *protrusum* (39.09 Mb) [28] and *C*. *dendroides* (36.69 Mb) [29]. However, the genome of another species in the same genus, *H*. *perniciosus* (44.0 Mb) [50], is larger and contains a higher proportion of repeated DNA sequences (25.27%). These findings suggest that genome size and repeat sequences can vary across species within the same genus, which could have implications for the evolution and ecology of these fungi [68].

During the early stages of infection by pathogenic fungi, the use of cell wall degrading enzymes to destroy the host cell wall is crucial for successful host infection [69]. The number of CAZymes in the genome of *C. mycophilum* is 499, which is higher than that of *C. protrusum* (412) [28] and *C. dendroides* (327) [29], possibly contributing to *C*. *mycophilum’s* ability to infect more hosts. The genome of *C. mycophilum* contains 30 GH18 genes, the most abundant type of the GH family. This family of genes produces a chitinase-like protein that aids in chitin degradation [70]. Since the mushroom cell wall primarily consists of chitin, we speculate that a large number of GH18 family members in the *C. mycophilum* genome are mainly utilized for the early infection stage of mushroom cell wall degradation, enabling *C. mycophilum* to invade mushroom cells and cause diseases.

Furthermore, the *C. mycophilum* genome also contains five GH75 genes that are associated with chitin degradation [71]. *C. mycophilum* genome contains 111 CAZymes genes encoding GT, the most among the family Hypocreaceae, with GT31 (15) being the most abundant. These GT gene families are mainly involved in chitin synthesis, cell wall biosynthesis, and glycosylation [50]. Therefore, the results suggest that CAZymes play a significant role in *C*. *mycophilum*’s ability to infect hosts.

Pathogenic fungi produce various secondary metabolites, including toxins, pigments, antibiotics, repellents, insecticides, anti-tumor, and cholesterol-lowering substances. In the case of *C*. *mycophilum* BJWN07, its secondary metabolite genes produce toxins, pigments, and compounds with potential resistance to harsh environmental conditions. One of these genes is *aur1*, which produces aurofusarin and possibly the red pigment in *C*. *mycophilum*. This pigment was first discovered in another fungus, *F*. *graminearum* [72]. There are also many unknown secondary metabolites in *C*. *mycophilum* BJWN07, suggesting that this strain has the potential to produce bioactive compounds.

When a pathogen infects a host, it produces a large number of virulence factors, including effector proteins, which play a crucial role in pathogenesis [73,74]. The genome of *C*. *mycophilum* BJWN07 contains 661 genes of secreted proteins, 1429 pathogen and host interaction-related genes, and 499 carbohydrate-active enzymes (CAZy) associated with the fungal host cell wall. All 1429 candidate disease-related proteins contained the PHI-base, which provides a valuable genetic resource for subsequent functional genomics research and in-depth investigation of disease-related proteins. These findings can aid in analyzing the pathogenic mechanism of *C*. *mycophilum* infection on mushrooms and developing effective prevention and control strategies for cobweb disease.

## Figures and Tables

**Figure 1 jof-09-00411-f001:**
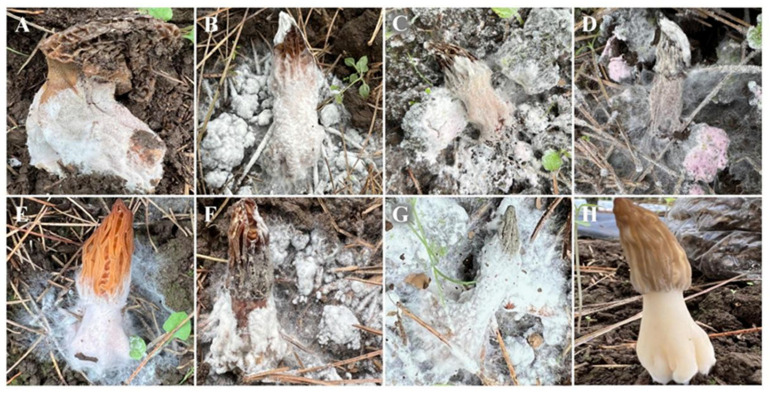
Field symptoms and pathogenicity of *M. sextelata* cobweb disease. (**A**–**D**) Field symptoms at different stages of the disease. (**A**): initial disease symptoms; (**B**,**C**): middle disease symptoms; (**D**): late disease symptoms. (**E**–**H**) Pathogenicity test where a spore suspension with a concentration of 5 × 10^6^ spores/mL was sprayed. (**E**): The symptoms of *M*. *sextelata* fruiting bodies 7 days after inoculation; (**F**): after 12 days of inoculation *M. sextelata* fruiting bodies stopped developing and suffered from soft rot; (**G**): after 15 d of inoculation, the fruiting bodies collapsed and died; (**H**): control, no disease symptoms appeared after 15 d of inoculation with sterile water.

**Figure 2 jof-09-00411-f002:**
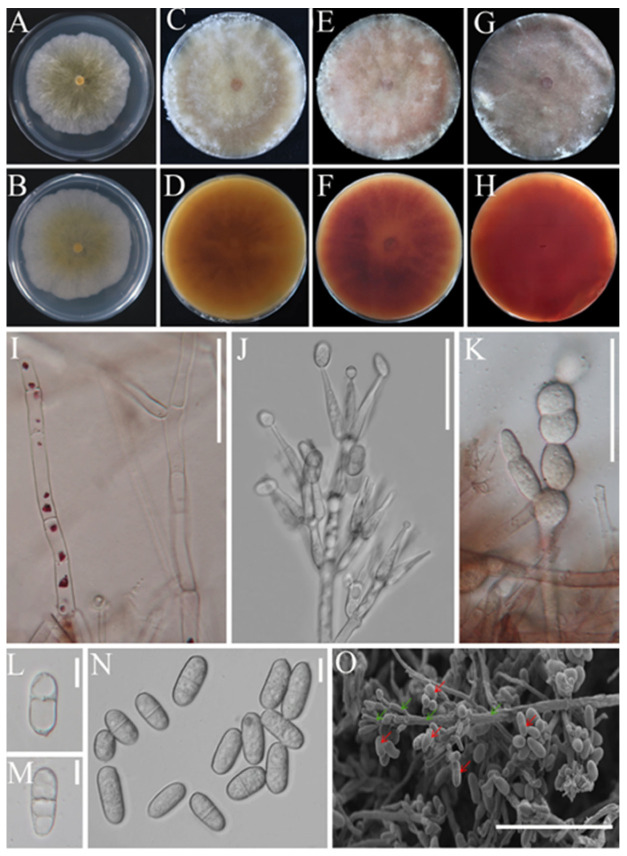
Colony and microscopic characteristics of *C*. *mycophilum*. (**A**,**B**): Depict the colony on PDA after 5 days and the reverse side, respectively; (**C**,**D**): depict the colony on PDA after 25 days and the reverse side, respectively; (**E**,**F**): depict the colony on PDA after 50 days and the reverse side, respectively; (**G**,**H**): depict the colony on PDA after 60 days and the reverse side, respectively. (**I**,**J**): Conidiophores, bar = 50 μm; (**K**): chlamydospores, bar = 50 μm; (**L**–**N**): conidiophores, bar = 10 μm; (**O**): conidiophores (green arrow) and conidiophores (red arrow) under the scanning electron microscope, bar = 100 μm.

**Figure 3 jof-09-00411-f003:**
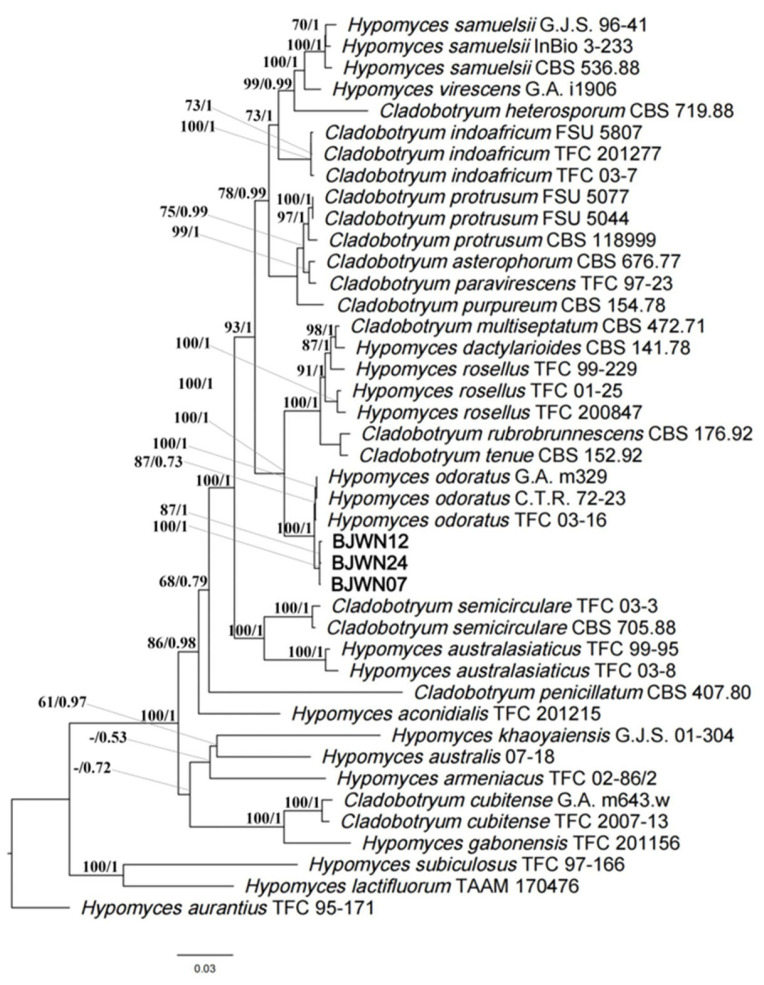
Phylogenetic tree based on rDNA ITS regions and partial sequences of TEF1 and RPB2 genes sequences for our three strains and selected reference isolates retrieved from GenBank of *Hypomyces*/*Cladobotryum*. Maximum likelihood (ML) values > 50% and Bayesian inference (BI) values > 0.50 are shown next to topological nodes and separated by “/”. Bootstrap values < 50% and BI values < 0.50 are labeled with “-”. *H. aurantius* was used as an outgroup. Newly generated sequences are indicated in bold.

**Figure 4 jof-09-00411-f004:**
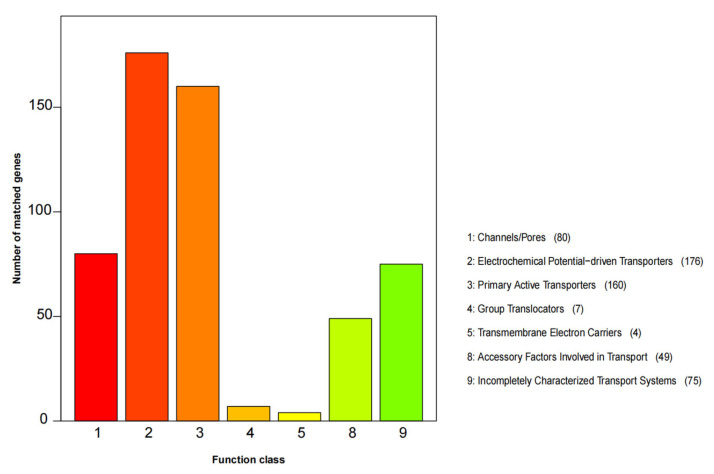
TCDB function classification of *C. mycophilum* BJWN07.

**Figure 5 jof-09-00411-f005:**
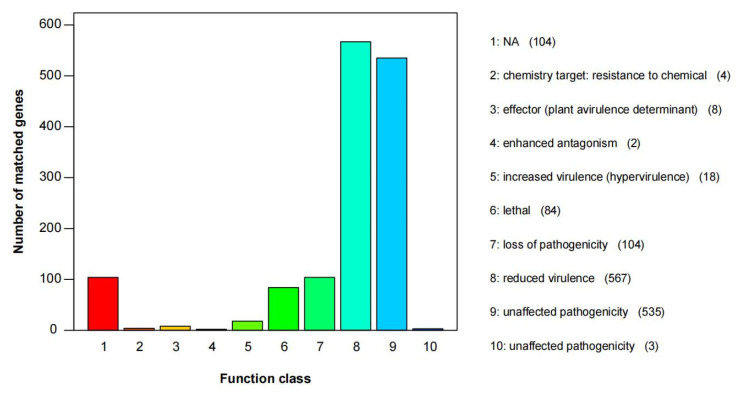
PHI functional annotation of *C. mycophilum* BJWN07.

**Figure 6 jof-09-00411-f006:**
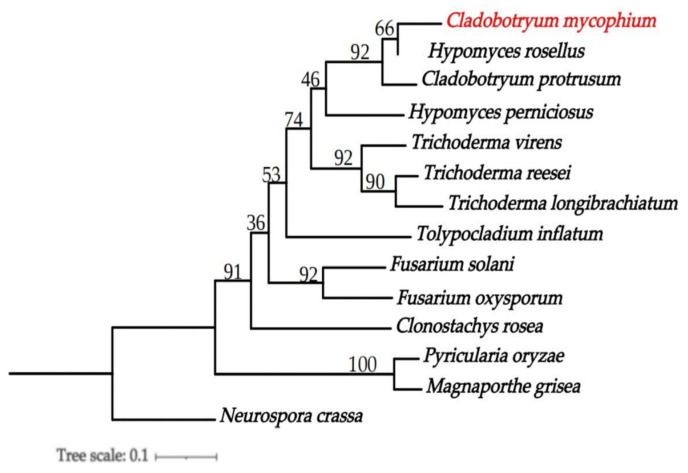
Phylogenetic tree of *C*. *mycophilum* and 13 other fungal species. Maximum likelihood (ML) values > 50% were placed close to topological nodes. The tree was rooted to *N. crassa*. The newly sequenced genome is indicated in red.

**Table 1 jof-09-00411-t001:** Sequences and GenBank accession numbers of *Cladobotryum* and *Hypomyces* isolates used in the phylogenetic analyses.

Species	Strain	Genbank Accession Numbers		
		ITS	TEF	RPB2
*C. asterophorum*	CBS 676.77	FN859395	FN868712	FN868649
*C. cubitense*	G.A. m643.w	FN859397	FN868714	FN868651
	TFC 2007-13	AM779857	FN868715	FN868652
*C. heterosporum*	CBS 719.88	FN859398	FN868716	FN868653
*C. indoafrum*	FSU 5807	FN859399	FN868717	FN868654
	TFC 03-7	FN859400	FN868718	FN868655
	TFC 201277	FN859401	FN868719	FN868656
*C. multiseptatum*	CBS 472.71	FN859405	FN868723	FN868659
*C. mycophilum*	**BJWN07**	OP714368	OP759638	OP718561
	**BJWN12**	OP714369	OP759639	OP718562
	**BJWN24**	OP714393	OP759640	OP718563
*C. paravirescens*	TFC 97-23	FN859406	FN868724	FN868660
*C. penicillatum*	CBS 407.80	FN859407	FN868725	FN868661
*C. protrusum*	CBS 118999	FN859408	FN868726	FN868662
	FSU 5044	FN859409	FN868727	FN868663
	FSU 5077	FN859410	FN868728	FN868664
*C. purpureum*	CBS 154.78	FN859415	FN868733	FN868669
*C. rubrobrunnescens*	CBS 176.92	FN859416	FN868734	FN868670
*C. semicirculare*	CBS 705.88	FN859417	FN868735	FN868671
	TFC 03-3	FN859418	FN868736	FN868672
*C. tenue*	CBS 152.92	FN859420	FN868738	FN868674
*H. aconidialis*	TFC 201215	FN859455	FN868774	FN868710
*H. armeniacus*	TFC 02-86/2	FN859424	FN868742	FN868678
*H. aurantius*	TFC 95-171	FN859425	FN868743	FN868679
*H. australasiaticus*	TFC 99-95	FN859427	FN868745	FN868680
	TFC 03-8	FN859428	FN868746	FN868681
*H. australis*	TFC 07-18	AM779860	FN868747	FN868682
*H. dactylarioides*	CBS 141.78	FN859429	FN868748	FN868683
*H. gabonensis*	TFC 201156	FN859430	FN868749	FN868684
*H. khaoyaiensis*	G.J.S. 01-304	FN859431	FN868750	FN868685
*H. lactifluorum*	TAAM 170476	FN859432	FN868751	EU710773
*H. odoratus*	C.T.R. 72-23	FN859433	FN868752	FN868687
	G.A. m329	FN859434	FN868753	FN868688
	TFC 03-16	FN859437	FN868756	FN868691
*H. rosellus*	TFC 99-229	FN859441	FN868759	FN868695
	TFC 01-25	FN859442	FN868760	FN868696
	TFC 200847	FN859438	FN868761	FN868692
*H. samuelsii*	CBS 536.88	FN859444	FN868763	FN868698
	G.J.S. 96-41	FN859448	FN868766	FN868702
	InBio 3-233	FN859450	FN868768	FN868704
*H. subiculosus*	TFC 97-166	FN859452	FN868770	EU710776
*H. virescens*	G.A. i1906	FN859454	FN868772	FN868708

Note: Isolates in bold are of the present study.

**Table 2 jof-09-00411-t002:** *Cladobotryum mycophilum* BJWN07 genome features.

Terms	BJWN07
The number of reads	300,757
Data size (bp)	5,272,956,422
Minimum sequencing read length (bp)	78
N50 Contig Length (bp)	18,317
Maximum sequencing read length	49,868
Genome size (Mb)	38.56
Number of contigs	10
GC content	47.84%
Coverage	130×
Number of coding genes	8428
The number of RNAs	330

**Table 3 jof-09-00411-t003:** Number of genes annotated in each database for the coding gene.

Database Used for Gene/Protein Annotation	Number of Genes
Nr	7766
GO	5683
KEGG	7447
KOG	1964
Pfam	5683
SwissProt	3049
TCDB	551
CAZy	499
Secretory_Protein	661
P450	155
PHI	1429
DFVF	443

**Table 4 jof-09-00411-t004:** Carbohydrate-active enzyme annotation results of *C. mycophilum* BJWN07.

Classification	Number
Carbohydrate-binding molecule (CBM)	50
Carbohydrate Esterase (CE)	28
Glycoside hydrolases (GHs)	249
Glycosyltransferases (GTs)	111
Polysaccharide lyases (PLs)	9
Auxiliary activities (AA)	52
Total	499

**Table 5 jof-09-00411-t005:** Secondary metabolic gene cluster and gene number statistics.

Clusters	Clusters_Number	Gene_Number
T1PKS	23	229
siderophore	1	2
NRPS	18	175
T1PKS, terpene	2	25
NRPS-like, T1PKS	2	21
NRPS, NRPS-like, T1PKS	4	85
NRPS, T1PKS	8	79
NRPS, NRPS-like, T1PKS, indole, terpene	1	29
NRPS-like	7	69
terpene	12	59
Total	78	773

## Data Availability

This paper’s genome sequence data and assembly are linked with NCBI BioProject: PRJNA917475 and BioSample: SUB12509998.

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
