# Peer review of "Characterization and Genome Analysis of Cladobotryum mycophilum, the Causal Agent of Cobweb Disease of Morchella sextelata in China"

_jof, 2023, doi:10.3390/jof9040411_

Round 1

Reviewer 1 Report

Well written and excellently presented. I enjoyed the article.

Reviewer 2 Report

The idea of the paper is nice, but the structure and quality of the paper needs major improvement. I recommend major revision. I recommend getting a native English speaker to check the paper as the language needs major improvement. Some of the major comments are:

1. The current title is too long and should be improved

2. Sentence at line 20-24 needs rewriting 

3. Many sentences are too long and needs rewriting

4. Why was a spore suspension with a concentration of 5×106 spores / mL used? This is very high and would likely result in disease in any plant. The author should mention details of control in the method

5. Line 181 needs correcting

6. The paragraph under section 3.1 is not the result and should be moved to another section

7. Please change Koch's rule to Koch postulate

8. The result section needs rearranging as per the method, for example you should mention the morphology of the species then pathogenicity test

9. The phylogenetic tree needs to be based on multi gene, not just ITS region for an accurate identification

10. I also suggest adding a taxa table for the gene regions of the isolates used for the analyses for future reference

Reviewer 3 Report

Morchella sextelata is a nutritious and tasty edible mushroom, and sheep's maw cultivation and growing has greatly expanded the edible mushroom industry. This manuscript presents for the first time the genome of Cladobotryum mycophilum, a pathogenic fungus of cobweb disease, and screens for proteins associated with its cure based on genomic information. These results provide a reference for understanding the pathogenic mechanism of cobweb disease and developing prevention and control strategies. This work has important theoretical and applied value, and is in line with the scope of this journal.

Main concerns

1.    Figures 4-7 is a schematic representation of the primary data provided by the sequencing company and is optional for the text.

2.    For the candidate causative genes analysed, it would have been more convincing to validate these results with the help of the transcriptome.

Minor errors

1. Line 46, “…, has become…”, grammatical mistake.

2. Line 48-52, “However, …industry”, grammatical mistake.

3. Line 57-61, “Cobweb disease is …”, grammatical mistake.

4. Line 63, “infects”

5. Line 66, Delete “the”

6. Line 70, “diseases”

7. Line 198-199, “After…” grammatical mistake.

8. Line 261, The GenBank accession numbers of EF-1α (OP759638, OP759639, and OP759640) and RPB2 (OP718561, OP718562, and OP718563) are not available in NCBI.

9. Line 318, “A total of 8482 protein-coding genes…”, not consistent with Table 1.

10. Line 363-366, The numbers are inconsistent with that in the figure 6. Check them carefully.

11. Line 398-399, “The top ….” The numbers are inconsistent with that in the figure 8. Check them carefully.

12. Line 426, What does “T1PKSe” means?

13. Line 440-443, “Proteins….” The numbers are inconsistent with that in the figure 9. Check them carefully

14. Line 457-460, Latin name of the species should be italic. You need to abbreviate the genus name of species have been mentioned before in manuscript.

15. Line 456-460, “Phylogenetic analysis showed that we isolated C. mycophilum BJWN07, the pathogen…….”, grammatical mistake.

16. Line 485, “incidence of the cobweb disease was 5% to 60%”, The number range is too broad. What is the median?

17. Line 507-509, What does this sentence mean?

18. Line 534, “aur1” should be italic if it’s a gene.

19. Line 578, “Morchella Angusticepes”, line 582, “Morchella Mycelium” should be italic.

19. The supporting information is not available. Check if that has been uploaded.

20. Most important, the genome file should be deposited in online databank like NCBI and make sure that the file could be accessed in public.

Round 2

Reviewer 2 Report

The paper can be accepted